# Vaginal microbiome *Lactobacillus crispatus* is heritable among European American women

Michelle L. Wright [1,2,10], Jennifer M. Fettweis[3,4,5,10], Lindon J. Eaves[6], Judy L. Silberg[6,7], Michael C. Neale [6], Myrna G. Serrano [3,5], Nicole R. Jimenez[3,5], Elizabeth Prom-Wormley[8], Philippe H. Girerd[4,5], Joseph F. Borzelleca Jr.[4], Kimberly K. Jefferson[3,5], Jerome F. Strauss III[4,5], Timothy P. York [4,6,11✉] & Gregory A. Buck [3,5,9,11]

The diversity and dominant bacterial taxa in the vagina are reported to be influenced by multiple intrinsic and extrinsic factors, including but not limited to pregnancy, contraceptive use, pathogenic states, socioeconomic status, and ancestry. However, the extent to which host genetic factors influence variation in the vaginal microbiota is unclear. We used a biometrical genetic approach to determine whether host genetic factors contribute to inter-individual differences in taxa from a sample of 332 twins who self-identified as being of African (44 pairs) or European ancestry (122 pairs). *Lactobacillus crispatus*, a major determinant of vaginal health, was identified as heritable among European American women (narrow-sense heritability = 34.7%, P-value = 0.018). Heritability of *L. crispatus* is consistent with the reduced prevalence of adverse reproductive disorders, including bacterial vaginosis and preterm birth, among women of European ancestry.

[1] School of Nursing, The University of Texas at Austin, Austin, TX, USA. [2] Department of Women's Health, Dell Medical School, The University of Texas at Austin, Austin, TX, USA. [3] Department of Microbiology and Immunology, School of Medicine, Virginia Commonwealth University, Richmond, VA, USA. [4] Department of Obstetrics and Gynecology, School of Medicine, Virginia Commonwealth University, Richmond, VA, USA. [5] Center for Microbiome Engineering and Data Analysis, Virginia Commonwealth University, Richmond, VA, USA. [6] Department of Human and Molecular Genetics, School of Medicine, Virginia Commonwealth University, Richmond, VA, USA. [7] Mid-Atlantic Twin Registry, Virginia Commonwealth University, Richmond, VA, USA. [8] Family Medicine and Population Health, Division of Epidemiology, Virginia Commonwealth University, Richmond, VA, USA. [9] Department of Computer Science, School of Engineering, Virginia Commonwealth University, Richmond, VA, USA. [10]These authors contributed equally: Michelle L. Wright, Jennifer M. Fettweis [12]These authors jointly supervised this work: Timothy P. York, Gregory A. Buck. ✉email: tpyork@vcu.edu

Differences in microbial diversity and predominant bacteria of the vaginal microbiome have been reported to be related to pregnancy[1], vaginal infections[2,3], contraceptive use[4], preterm birth[5–8], menopause[9], and ethnic/racial background[10–13]. Women harboring more complex vaginal microbiomes are more likely to have bacterial vaginosis, higher susceptibility to sexually transmitted diseases, pelvic inflammatory disease, and premature birth[14]. Yet, the contribution of host genetic factors to shaping the vaginal microbiota is largely unknown.

Overall, studies published to date suggest there are not broad host genetic influences, but rather genetic factors contribute to presence of specific microbes associated with clinical conditions. Twin studies of gut microbiome composition in monozygotic (MZ) and dizygotic (DZ) twin pairs show[15,16] that some bacteria appear to be heritable and associate with a clinical phenotype[15]. One study reports that the vaginal microbiomes of MZ twins (n = 26, 13 pairs) are more similar to each other than to their mothers (n = 8) or sisters (n = 8)[9]. Yet, this study does not adequately distinguish host genetic from other shared familial influences. Another study evaluating the vaginal microbiome among Korean twins and selected relatives (N = 542, 111 MZ and 28 DZ pairs), reports *Prevotella* as the most heritable bacterial taxon in the vaginal microbiome[17]. However, most of the taxa are reported at the genus level. Given the importance of species-level differences to women's health outcomes, further studies with improved resolution among more diverse cohorts are warranted.

The estimation of a genetic contribution to inter-individual differences in phenotypic trait measurements was proposed over a century ago[18]. The subsequent development of methods to partition a trait of interest into separate genetic and environmental contributions using genetically informative twin and family samples remains an important tool in the genomics era[19]. Heritability is a technical term that refers to the proportion of phenotypic variance of a trait measured in a population that can be accounted for by genetic sources. A meta-analysis of over 17,804 human traits from 2748 twin studies report an average heritability of 49%, and a subset of reproductive traits at an average heritability of 31%[20]. The utility of this summary ratio of genetic variance to the total phenotypic variance not only expresses the extent of genetic influence but also allows for meaningful comparisons across traits. An accurate representation of the genetic architecture of species-level vaginal microbiota requires heritability estimates across different populations.

Evaluating host genetic and environmental contributions to the vaginal microbiome composition has unique challenges compared to other microbiome sites due to the sparsity of data and relatively few numbers of taxa present across all women within the vagina. Vaginal microbiomes of ~60–90% of reproductive aged women are predominantly composed of *Lactobacillus* species[10,11,13]. High proportions of specific species within vaginal microbial communities are associated with vaginal health (e.g., *Lactobacillus crispatus*) and adverse clinical conditions (e.g., *Gardnerella vaginalis* with bacterial vaginosis). Previous studies consistently report different proportions of predominant taxa by self-reported ancestry[10,11,13]. For example, *L. crispatus* is more prevalent among women of European ancestry, whereas *Lactobacillus iners* is the most prevalent *Lactobacillus* species among women of African ancestry[10,11].

The goal of this study was to estimate the contribution of host genetic factors to species-level variation in microbial taxa of the vagina using a biometrical genetic approach. Differences in these estimates by self-reported ancestry were considered due to consistently reported heterogeneity in vaginal microbiome composition between ancestry groups, including our previous studies[10,12].

**Table 1 Reproductive health characteristics of twin participants by self-reported ancestry.**

| Variable | European ancestry (N = 244) | African ancestry (N = 88) | P value |
|---|---|---|---|
| **Zygosity (%)** | | | |
| MZ | 168 (69) | 48 (55) | |
| DZ | 76 (31) | 40 (45) | 0.0224 |
| **BMI** | | | |
| Minimum | 18.00 | 17.00 | |
| Median (IQR) | 25.10 (21.00, 31.02) | 30.90 (25.30, 36.00) | |
| Mean (sd) | 26.84 ± 6.74 | 30.95 ± 7.02 | 0 |
| Maximum | 50.00 | 49.00 | |
| Missing | 32/244 (13) | 11/88 (12) | |
| **Age** | | | |
| Minimum | 19 | 18 | |
| Median (IQR) | 36.00 (29.00, 49.00) | 36.50 (29.00, 51.75) | |
| Mean (sd) | 39.58 ± 13.90 | 38.98 ± 13.40 | 0.7199 |
| Maximum | 78 | 65 | |
| **Bacterial vaginosis (%)** | | | |
| No | 213 (89) | 65 (77) | |
| Yes | 19 (8) | 18 (21) | |
| Not sure | 6 (3) | 1 (1) | |
| Missing | 6/244 (2) | 4/88 (5) | 0.0021 |
| **Pregnant status (%)** | | | |
| No | 231 (96) | 85 (97) | |
| Yes | 4 (2) | 1 (1) | |
| Not sure | 6 (2) | 2 (2) | |
| Missiing | 3/244 (1) | 0/88 (0) | 1 |
| **Nulliparous (%)** | | | |
| No | 164 (69) | 62 (70) | |
| Yes | 75 (31) | 26 (30) | |
| Missing | 5/244 (2) | 0/88 (0) | 0.8543 |
| **Hormone therapy (%)** | | | |
| No | 64 (78) | 34 (92) | |
| Yes | 18 (22) | 3 (8) | |
| Missing | 162/244 (66) | 51/88 (58) | 0.1155 |
| **Hormonal birth control (%)** | | | |
| No | 162 (75) | 61 (77) | |
| Yes | 54 (25) | 18 (23) | |
| Missing | 28/244 (11) | 9/88 (10) | 0.811 |
| **Sample pH (%)** | | | |
| Minimum | 4.00 | 4.00 | |
| Median (IQR) | 4.50 (4.00, 5.50) | 4.70 (4.40, 5.50) | |
| Mean (sd) | 5.02 ± 1.13 | 5.08 ± 1.03 | 0.6369 |
| Maximum | 7.00 | 7.00 | |
| Missinig | 4/244 (2) | 0/88 (0) | |
| **Current smoking (%)** | | | |
| No | 142 (67) | 64 (83) | |
| Yes | 71 (33) | 13 (17) | |
| Missing | 31/244 (13) | 11/88 (12) | 0.0099 |

## Results

A total of 380 mid-vaginal wall swabs were obtained from self-identified MZ or DZ twin participants (Table 1). Thirty-four samples were collected from only one member of a twin pair and not included in these analyses. There were 332 twins of self-identified African (44 pairs) or European ancestry (122 pairs). There was a higher proportion of African American DZ than MZ twins (chi-square = 5.21, P value = 0.022). Participants ranged in age from 18 to 78 years old (median = 36). The median BMI of all participants was 27 (range = 17–50), and the mean was higher among African American participants (t-test = 4.45, P value < 0.001). In this sample African American participants were more often diagnosed with bacterial vaginosis (chi-square = 9.48, P value = 0.002), while more European

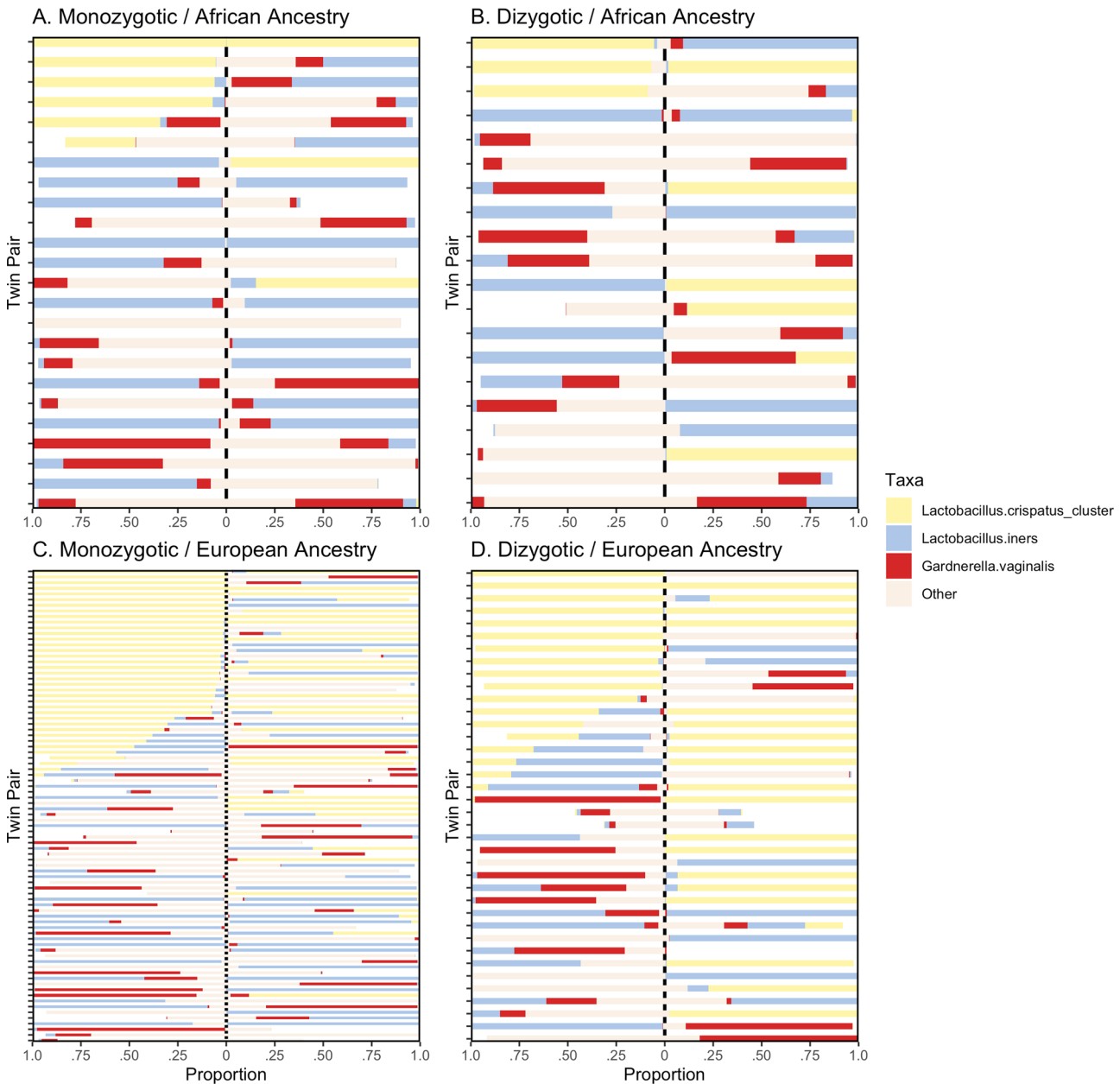

**Fig. 1 Taxa distribution across zygosity and ancestry.** Twin pairs are represented as one horizontal line within a pane with each member separated by the vertical dashed line. Monozygotic twin pairs are presented in the left panels, with dizygotic twin pairs on the right. The upper panels are twin pairs of African ancestry, and European ancestry twin pairs are the bottom two panels. Panels summarize taxa proportions for **A**. Monozygotic/African ancestry; **B**. Dizygotic/African ancestry; **C**. Monozygotic/Eurpean ancestry and; **D**. Dizygotic/European ancestry twin pairs.

American participants reported active smoking at the time of assessment (chi-square = 6.66, P value = 0.010).

After initial quality control screening of samples containing at least 5000 reads, 267 bacterial taxa were identified in at least one participant. Due to the sparsity of individual taxa in the vaginal microbiome data, we limited analyses to 3 taxa, *L. cripatus*, *L. iners*, and *G. vaginalis*, that were present in at least 20% of participants and a within participant proportion of at least 10% (Fig. 1). Biometrical analyses identified a single taxon, *L. crispatus* (heritability = 34.7%, P value = 0.018), as significant among women of self-reported European ancestry and did not identify any taxa as heritable for women of self-reported African ancestry (Table 2). *L. crispatus* is the taxon most commonly associated with vaginal health, particularly among women of European ancestry. The genetic contribution of almost 35% of variance in *L.*

*crispatus* abundance among women of European ancestry may partially explain why a higher prevalence of *L. crispatus* is consistently reported among this population. Heritability estimates remained constant after removing women older than the mean age of menopause onset (51 years) to assess whether the expected shift in vaginal pH influenced the genetic covariance of these taxa (Supplementary Table 1).

## Discussion

A landmark study from the Gordon laboratory demonstrated that gut microbiota from twin pairs discordant for obesity could be transplanted into gnotobiotic mice and differentially modulate metabolism[21], established the microbiome as an environmental factor that can have strong phenotypic effects. More recent studies

**Table 2 Heritability estimates for predominant taxa by self-reported ancestry.**

| Taxa | Ancestry | Component | Standardized estimate | Unstandardized estimate | 95% LI | 95% UI |
|------|----------|-----------|----------------------|------------------------|--------|--------|
| *Lactobacillus crispatus* | African | A | 0.1706 | 0.0455 | −0.0609 | 0.1630 |
| | | E | 0.8294 | 0.2213 | 0.1358 | 0.3735 |
| | European | A | 0.3473 | 0.1428 | 0.0583 | 0.2415 |
| | | E | 0.6527 | 0.2683 | 0.2022 | 0.3625 |
| *Lactobacillus iners* | African | A | 0.0705 | 0.0215 | −0.0981 | 0.1463 |
| | | E | 0.9295 | 0.2835 | 0.1788 | 0.4669 |
| | European | A | 0.0867 | 0.0217 | −0.0278 | 0.0738 |
| | | E | 0.9133 | 0.2292 | 0.1769 | 0.3019 |
| *Gardnerella vaginalis* | African | A | 0.1259 | 0.0116 | −0.0236 | 0.0506 |
| | | E | 0.8741 | 0.0805 | 0.0502 | 0.1331 |
| | European | A | −0.1654 | −0.0168 | −0.0396 | 0.0045 |
| | | E | 1.1654 | 0.1187 | 0.0914 | 0.1563 |

*LI* lower interval, *UI* upper interval, *A* additive genetic component, *E* unique environmental component.

of the gut microbiota have found that relatively few species exhibit strong heritability, with some heritability reported at higher taxonomic levels (e.g., Christensenellaceae family, *Bifidobacterium* abundance)[22,23]. Taxa found in the gut with the strongest heritability have been associated with clinically meaningful phenotypic differences, such as host metabolism[15]. Two genome-wide analyses related to microbiome composition using Human Microbiome Project data concluded that host genetic signature likely influences the overall composition of the oral and gut microbiome[22,23]. However, differences in approach and significance testing resulted in the lack of replication between studies of genetic associations related to the presence of specific taxa. One of the studies evaluated the relationship of host genetic factors and the vaginal microbiome ($N = 80$); but did not identify any statistically significant associations, likely due to the small sample size[23].

In the current study, we reported significant heritability estimates for *L. crispatus* among women of European ancestry, but not among women of African ancestry. The difference in heritability between self-identified ancestry groups in this case may be due to the increased prevalence of *L. crispatus* dominance among women of European ancestry[10,12]. Larger sample sizes will be needed to determine whether this taxon is also heritable among women of African ancestry given that prevalence of *L. crispatus* is less common[10–13,24]. Additionally, we did not identify significant heritability estimates for *L. iners* or *G. vaginalis* among women of European or African ancestry.

Two previous twin studies evaluated heritability of the vaginal microbiome among Korean women[9,17]. Lee et al. reported the vaginal microbiome of twins are more alike to each other than their mothers or sisters, but noted that differing menopause status could have contributed to the observed differences[9]. Heritability estimates for individual microbial taxa were not reported. In the second larger study that included 111 MZ and 28 DZ pairs, *Prevotella* was identified as the most heritable taxa (48.76%) followed by *L. crispatus* (36.9%) and *L. iners* (41.2%). Thus, *L. crispatus* has similar heritability estimates among Korean and European American women. The study by Si and colleagues reported most heritability estimates at the genus level; other than *L. crispatus* and *L. iners*, species-level differences in the vaginal microbiome among their cohort were not reported[17].

Differences in vaginal microbiome composition by ancestry are one of the most reproducible signatures reported in microbiome literature. *L. crispatus* dominance is more common among women of European ancestry[10]. Women of African ancestry are less likely to have *L. crispatus* dominance and more likely to have *L. iners* or diverse microbial communities[10,14]. These differences are observed across geographic locations and populations[3,5,13],

suggesting there may be a heritable component contributing to the compositional differences of the vaginal microbiome. Furthermore, the association of *L. crispatus* with lower risk for preterm birth is one of the most consistent, reproducible microbial signatures identified to date[5,7,25,26]. Understanding what contributes to variation in the microbiome is clinically important because *L. crispatus* dominance is associated with lower risk for vaginal infections and preterm birth[10–12]. Additionally, diverse vaginal microbial communities have been associated with birth complications and higher risk for sexually transmitted infections, including HIV infection and tansmission[5,14]. However, the lowered risk of preterm birth associated with *L. crispatus* prevalence is only consistently reported among women of European ancestry, which may be partially due to the lower prevalence of *L. crispatus* among women of African ancestry. This is the only study to date that has attempted to estimate the genetic contribution to vaginal microbiome profiles among African American and European American women. Our findings raise important questions regarding how host genetic factors influence the establishment of the vaginal microbiome and subsequently contribute to women's health. Future studies should be directed to the identification of specific genetic loci that contribute to the heritability of *L. crispatus*.

Understanding heritable differences in host susceptibility will be critical for precision-based approaches of microbiome modulation to improve health outcomes. *L. crispatus* is one of the most promising species being tested as a probiotic to improve and maintain vaginal health[27]. If *L. crispatus* is indeed heritable, but only among some ancestry groups, this may have implications for probiotic effectiveness in improving vaginal health or birth outcomes in diverse populations. Future work investigating efficacy of *L. crispatus* probiotics need to engage women across ancestry groups, and may need to consider alternate approaches among women without previous *L. crispatus* predominance (e.g., longer duration or different probiotic therapy). To date, published research on specific taxa of the vaginal microbiome has largely focused on common species, such as those classified as *Lactobacillus* or *Gardnerella*. Understanding how relatively unstudied, less prevalent, microbes contribute to vaginal health and birth outcomes may be necessary to solve persistent clinical puzzles, such as recurrence of bacterial vaginosis. We will likely need a variety of approaches for microbiome modulation if host genetic factors contribute to vaginal microbiome signatures.

In summary, our study suggests *L. crispatus*, is a heritable taxon present in the vaginal microbiome. Therefore, heritability may explain, in part, why *L. crispatus* is most prevalent among women of European ancestry in previous vaginal microbiome studies.

## Methods

Adult female twin pairs were recruited from the Mid-Atlantic Twin Registry under IRB protocol HM12169 at Virginia Commonwealth University. Detailed sample collection and data processing methods have been described elsewhere[10,28]. Briefly, metadata were collected from 173 (112 MZ; 61 DZ) female twin pairs over the age of 18, 166 pairs self-identified as African or European ancestry. Swabs from the mid-vaginal wall were collected by a physician during a speculum exam using CultureSwab EZ (Becton Dickenson) and DNA was extracted within 4 h of collection using the Powersoil kit (MoBio). V1-V3 regions of bacterial 16S rRNA were amplified and sequenced using the Roche 454 GS FLX Titanium platform. Amplification primers specifically designed to interrogate the vaginal microbiome using a (4:1) mixture of the forward primers Fwd-P1 (5′ - CCATCTCATCCCTGCGTGTCTCCGACTCAG BBBBBB AGAGTTYGATYMTGGCTYAG) and Fwd-P2 (5′ - CCATCTCATCCCT GCGTGTCTCCGACTCAG BBBBBB AGARTTTGATCYTGGTTCAG) and the reverse primer Rev1B (5′ – CCTATCCCCTGTGTGCCTTGGCAGTCTCAG ATTACCGCGGCTGCTGG) as previously described[10,28]. The analysis pipeline controls for detection (> 5,000 reads), quality of base reads, and assigns taxonomy with minimum bootstrap confidence of 80% using STIRRUPS (Species-level Taxon Identification of rDNA Reads using USEARCH Pipeline Strategy) using database version 09/27/11[28]. Allelic tests using the Profiler Plus and Cofiler kits were performed to determine the zygosity status for 14 pairs of twins with missing relationship information. The zygosity status of a random sample of 20 pairs of known zygosity (10 MZ and 10 DZ pairs) were evaluated to validate self-reported measures.

**Data analysis**. Briefly, genetic and environment contributions to variance in taxa abundance can be derived from the observation that MZ and DZ twins share differing proportions of genetic material and also share environmental exposures as a result of being raised together. Differences within MZ twins of the same pair are assumed to be due to unique environmental influences, which also contain measurement error in the modeling framework. Since members of a DZ twin pair share, on average, only one-half their genes observed differences can be attributed to their genes not shared, in addition to non-shared environmental influences. Thus, taxa showing stronger correlations within MZ pairs compared to DZ pairs can be attributed to genetic factors. The twin study provides a methodological control well-suited for vaginal microbiome studies where vertical transmission may occur since both members of MZ and DZ twin pairs could, in practice, be exposed to the same microbiota during delivery. Exposures shared by both members of a twin pair would be reflected in the estimates of the common environment (C) as opposed to additive genetic sources (A). In this study, the OpenMx R package was used to solve a system of linear equations using maximum likelihood methods to estimate the contribution of genetic (A), common environment (C) and unique environmental (E) sources[29]. The arcsine square root transformation of taxa proportions was used for all heritability analyses and was estimated for each taxa and self-identified ancestry group separately. In order to ensure sufficient covariance coverage, heritability was estimated only for taxa that were present in at least 20% of subjects and a within subject proportion of at least 10%. The contribution of individual genetic and environmental parameters to the covariance of taxa was assessed by dropping each in turn from the model and registering the decline in the fit of sub-models by the likelihood ratio chi-square test and change in the Akaike Information Criterion. In order to limit type I errors, an alpha level of 5% was used for all statistical tests. Nested model and fit statistics are provided in Supplementary Table 2a–f for each taxa by ancestry. An expanded list of 32 taxa present in at least 2.5% of subjects and a within subject proportion of at least 1% can be found in Supplementary Tables 3a–d insofar as this can be helpful for comparisons in future studies. The code used to compute heritability is available at https://github.com/tpyork/twin-microbiome.

**Reporting summary**. Further information on research design is available in the Nature Research Reporting Summary linked to this article.

## Data availability

Data used for this study has been uploaded to dbGAP under Study Accession number phs000256. Any remaining information can be obtained from the corresponding author upon reasonable request.

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

## Acknowledgements

We acknowledge the twins who contributed to the study with appreciation; we thank the research coordinators and sample processors who assisted with sample collection. The study was funded by NIH grant nos. UH3AI083263 and U54HD080784, with funds from the Common Fund, the National Center for Complementary and Integrative Health, and the Office of Research on Women's Health. Additional support included a GAPPS BMGF PPB grant to G.A.B. and J.M.F., and NIH grants R21HD092965 to J.M.F. and E. Wickham, K01NR017903 to M.L.W., and 1R01HD092415 to G.A.B. N.R.J. was supported by grant no. R25GM090084 for the VCU Initiative for Maximizing Student Development program. L.P.W was supported by grant T32MH020030. Sequence analysis was performed in the Nucleic Acids Research Facilities at VCU and analysis was performed with servers provided by the Center for High Performance Computing at VCU. Twin pairs were recruited through the Mid-Atlantic Twin Registry at VCU.

## Author contributions

M.L.W., J.M.F, L.J.E, K.K.J, J.F.S, T.P.Y, and G.A.B. designed the study. J.M.F, P.H.G., J.F.B and J.F.S comprised the clinical team; J.L.S. and E.P.W led the ascertainment of twin pairs. J.M.F., M.G.S, K.K.J. and G.A.B performed microbiome sequencing and classification. M.L.W. and T.P.Y. performed all analyses with input from J.M.F., L.J.E, E.P.W. and M.C.N. The manuscript was drafted by M.L.W, J.M.F., N.R.J and T.P.Y. and critically reviewed and approved by all authors.

## Competing interests

The authors declare no competing interests.
