## [Peer Review File · Communications Biology]

Reviewers' Comments:

Reviewer #1:

Remarks to the Author:

In the manuscript "Vaginal microbiome *Lactobacillus crispatus* is heritable among women of European ancestry" Wright et al. apply "biometrical analyses" to the comparison of bacterial taxonomic microbiota compositions of 322 twin pairs, including 122 twins of European and 44 of African ancestry, to identify heritable microbial taxa. Due to inter-individual differences between women and the sparsity of data (taxa not being present in many samples), the authors limit their analysis to two lactobacilli species, *L. crispatus*, *L. iners*, and *G. vaginalis*. The authors identify *L. crispatus* as the only heritable taxon and only in women of European ancestry. There are no other predicted heritable taxa in either women of European or African ancestry.

The identification and comparison of microbiota heritability contributions, as opposed to vertical microbiota transmission or microbiota acquisition from environmental sources to microbiome composition and function has been the subject of many studies, which have so far mostly focused on the gut and recently attributed larger contributions to environmental exposure rather than human genetics. Therefore, it is relevant and interesting to address this research question for the vaginal microbiome and the authors have assembled an impressive cohort and dataset.

However, the presented analysis is not very convincing and the results are somewhat disappointing, as (i) it is difficult to evaluate the robustness of the applied statistical model without being an expert in statistics (for example, how different exactly are the predictions of heritability for mono- and dizygotic twins?); (ii) the underlying 16S rRNA dataset seems not very suitable to reliably identify heritability (the authors explain that only three species pass the filtering threshold); (iii) the presented findings are not confirmed or expanded with any other experimental or bioinformatic data (host genetics, phylogenetic analyses of host or microbial genomes or specific genetic loci, etc.); and (iv) the authors present only very limited evidence for heritability (single species, only women of European ancestry).

Major comments:

1. Heritability prediction

I know it is difficult, but since this is such a crucial component of their manuscript, the authors should better document and explain how heritability was determined.

- How does the underlying model work?
- What does a heritability score of 34.7% actually mean?
- Which other measures or references can the authors cite to document the reliability of their predictions?
- Can the authors include a positive and negative control of other datasets to support the reliability?
- Are there any other public datasets that the authors could use to confirm their findings (e.g., from mothers and their daughters)?
- The fact that the only identified heritable taxon is also the bacterial species with the highest prevalence and relative abundance across all samples suggests a potential dependency of the analysis on these taxon features. Vaginal birth, which has already been demonstrated to result in vertical transmission of gut and vaginal microbes from the mother to the infant gut, could also result in the transfer of the most prevalent and abundant bacterial taxon, *L. crispatus*, to the vaginal microbiota of female infants.

2. Methods description

The authors should include more basic methods information into the main manuscript (l. 98), such as

the type of quality control screening and the type of sequence data (16S rRNA amplicon not metagenomic sequence reads).

The authors should also mention that species assignments can be achieved with 16S rRNA amplicon sequence data for the vaginal microbiota and its species (can they?). - This is controversial, at least for other body sites and taxa!

Other comments:

1. "Women of African ancestry tend to harbor more complex vaginal microbiomes that are often associated with bacterial vaginosis, higher susceptibility to sexually transmitted diseases, pelvic inflammatory disease, and premature birth" (l. 56) - The authors should split this sentence and rephrase. It might be technically true, but it sounds terrible to associate African ancestry with bacterial vaginosis, sexually transmitted diseases, premature birth, etc.

2. Throughout the manuscript (abstract + main text): "There were 322 twins of self-identified African (44 pairs) or European ancestry (122 pairs)."  $2 \times (122 + 44) = 332$

3. "Our study has not identified any specific genetic loci that contribute to the heritability of *L. crispatus*." (l. 152) - This is misleading, as the analysis of host genetics was not part of the study.

4. "...published research... HAS largely focused..." (l. 164)

5. Table 1

- explain numbers in parentheses

- "Pregnant status" - Should the groups of "Not sure" and "Unknown" be combined?

Reviewer #2:

Remarks to the Author:

The authors tackle the interesting question of whether features of the vaginal microbiome are heritable. They conduct a twin study with twins of European ancestry and twins of African ancestry. Ancestry was self-reported. They then looked at three common vaginal taxa and asked if there was evidence of heritability and find that *L. crispatus* is heritable among women of European ancestry. I detail some concerns about the approach and conclusions below.

Doesn't age also influence vaginal microbiome composition? The subjects are 18 to 78 years old. Could this influence results?

How strong is the heritability score? I see that it's similar to something found in a Korean study (were the methods the same?) but I didn't get a sense from the paper of either how strong or how general we expect this observation to be.

Could the "lack of heritable taxa" described be simply explained because you only looked at 3 taxa? I was not clear on why the additional taxa that are described, were not used, or, if they were used, why this was not discussed in more detail?

There are no quantitative "host genetic factors" reported, there's just self reported ancestry? This makes me confused as to what this study is adding to prior observations in many other studies that *L. crispatus* is common in women of European descent. "The difference in heritability between self-identified ancestry groups in this case may be due to the increased prevalence of *L. crispatus* dominance among women of European ancestry" this is a statement from the paper so the authors seem to agree.

The heritability model seems to be used out of the box with default parameters but is not clearly described enough for a reader to reproduce the results. This is not acceptable. What are the different A C E model parameters? How were these decided upon? What are the biological rationales for such models? How are they generally used with microbiome data? What are the particular features of microbiome data that make these models appropriate or inappropriate?

In sum, I cannot explain clearly what additional contribution this study makes to prior studies that observe different distributions of *L. crispatus* in women of different ancestries. This, and other points above, would need to be addressed for the paper to be acceptable for publication.

Reviewer #1 (Remarks to the Author):

In the manuscript "Vaginal microbiome *Lactobacillus crispatus* is heritable among women of European ancestry" Wright et al. apply "biometrical analyses" to the comparison of bacterial taxonomic microbiota compositions of 322 twin pairs, including 122 twins of European and 44 of African ancestry, to identify heritable microbial taxa. Due to inter-individual differences between women and the sparsity of data (taxa not being present in many samples), the authors limit their analysis to two lactobacilli species, *L. crispatus*, *L. iners*, and *G. vaginalis*. The authors identify *L. crispatus* as the only heritable taxon and only in women of European ancestry. There are no other predicted heritable taxa in either women of European or African ancestry.

The identification and comparison of microbiota heritability contributions, as opposed to vertical microbiota transmission or microbiota acquisition from environmental sources to microbiome composition and function has been the subject of many studies, which have so far mostly focused on the gut and recently attributed larger contributions to environmental exposure rather than human genetics. Therefore, it is relevant and interesting to address this research question for the vaginal microbiome and the authors have assembled an impressive cohort and dataset.

However, the presented analysis is not very convincing and the results are somewhat disappointing, as (i) it is difficult to evaluate the robustness of the applied statistical model without being an expert in statistics (for example, how different exactly are the predictions of heritability for mono- and dizygotic twins?);

RESPONSE: The twin method has been used in genetic epidemiology studies for over 50 years and is the standard approach for estimating trait heritability on thousands of human traits. In this case we estimated heritabilities derived from measured microbiome proportions within and between twin pairs (page 7, line 245). For those less familiar with this approach we provide a rationale (new paragraph on page 2, lines 83-92) and details for the overall approach on page 7, lines 239-251 which includes new text. The biological rationale for the twin model is provided on page 8, lines 241-251. Some confusion by the reviewer might stem from the incorrect notion that there is a separate heritability by twin type (i.e., MZ/DZ). A full explanation of the biometrical twin model is beyond the scope of the current manuscript but we do commit significant space to describe the motivation of the approach and how other groups have used these methods to estimate the heritability of taxa in both the gut and vagina (page 2, lines 62-81; page 4, lines 132-152; page 5, lines 162-170). We make a point to indicate that the magnitude of our heritability results are consistent in what has been found in other twin and family studies. Importantly, as other studies from the gut microbiome have found, the taxa identified with the highest heritability estimates are those with clinical meaningful phenotypic associations (page 4, line 130).

(ii) the underlying 16S rRNA dataset seems not very suitable to reliably identify heritability (the authors explain that only three species pass the filtering threshold);

RESPONSE: The average sequencing depth of the 16S rRNA data was ~30,000 reads per sample, which is deep enough to sample both dominant and more minor taxa in the vaginal microbial community. The database used for taxonomic classification (PMID: 23282177) has been shown to map ~95% of minimally filtered reads. A total of 32 taxa that passed initial filtering for heritability analysis. This can be attributed to the structure of the vaginal microbiome composition, which is less diverse than many other niches of the human body, such as the GI tract.

(iii) the presented findings are not confirmed or expanded with any other experimental or bioinformatic data (host genetics, phylogenetic analyses of host or microbial genomes or specific genetic loci, etc.); and

RESPONSE: Understanding how host genetic factors (*i.e.*, heritability) explains variability in microbiome taxon abundance is the initial step in describing the genetic architecture of these traits. This type of genetic epidemiology study is essential to inform subsequent genetic studies (*e.g.*, the identification of specific genetic loci), yet only two have been performed to date, and both on Korean samples. There are no other experimental methods that can be used to confirm these estimates and so we make the point to report our results are consistent with other twin and family studies of host gut and vaginal microbiomes.

(iv) the authors present only very limited evidence for heritability (single species, only women of European ancestry).

RESPONSE: Our findings are consistent with the limited existing results that have shown the lack of global genetic influences across taxons while taxa that are estimated as heritable frequently are of clinical significance (page 2, line 62). It is important to provide our results from African American and European American samples that show *L. crispatus* as being highly heritable in the latter group since it has consistently been shown to be associated with vaginal infections and preterm birth. We discuss how our results are important for understanding the genetic architecture of these taxa and how they might inform precision-based approaches for microbiome modulation to improve health outcomes (page 6, line 194-214).

Major comments:

1. Heritability prediction

I know it is difficult, but since this is such a crucial component of their manuscript, the authors should better document and explain how heritability was determined.

- How does the underlying model work?

RESPONSE: We have provided an additional paragraph to provide more context to the twin method of estimating heritability (page 3, lines 83-92). The twin method has been used in genetic epidemiology studies for over 50 years and is the gold-standard approach for estimating trait heritability on thousands of human traits. In this case we estimated heritabilities derived

from measured microbiome proportions within and between twin pairs (page 8, line 245). We provide a technical and non-technical rationale and details of the approach beginning on page 8, lines 241-251. The biological rationale for the twin model is provided beginning on page 8, lines 241. The model parameters are described on page 8, line 251 and their relationship to the MZ and DZ covariance is mathematically defined in the code on lines 45-54. In order to promote replicability of our results we have provided a Github link for the code used to fit and test the models presented in this paper on page 8, line 264. All starting values and parameters are provided in the code to recreate our findings.

- What does a heritability score of 34.7% actually mean?

RESPONSE: We have clarified this term on page 4, line 126. Please refer also to our new paragraph that includes information on heritability context and expectations for these estimates (page 3, lines 83-92).

- Which other measures or references can the authors cite to document the reliability of their predictions?

RESPONSE: There are no such controls for the modern classical twin design reliably used in practice to estimate heritability for over 50 years (Martin and Eaves, 1977; PMID: 268313). It should be noted that the magnitude of our heritability findings are similar to previously published findings of human traits (average = 49%) including reproductive traits (average = 45%) (see: Polderman TJC, et al. Meta-analysis of the heritability of human traits based on fifty years of twin studies. Nat Genet. 2015;47:702–9). This information has been added (page 3, lines 83-92).

- Can the authors include a positive and negative control of other datasets to support the reliability?

RESPONSE: The strength of the heritability estimated for *L. crispatus* in European Americans was reported at 34.7% (Abstract and page 4, line 122). This estimate is similar to the value obtained from samples of Korean women (ie., 36.9%) which provides for a consistent estimate across at least two samples and thus our study would agree with expectations. We report on page 5, line 164-168 that the Korean sample used both MZ and DZ twin pairs similar to our study and using a similar sample size as ours.

- Are there any other public datasets that the authors could use to confirm their findings (e.g., from mothers and their daughters)?

RESPONSE: Unfortunately, besides those already mentioned in the manuscript, there exists no additional twin and family study with abundance levels on the vagina microbiome. We review these other findings on page 5, lines 162-170.

- The fact that the only identified heritable taxon is also the bacterial species with the highest prevalence and relative abundance across all samples suggests a potential dependency of the analysis on these taxon features.

RESPONSE: We agree and the sparse data problem in vaginal microbiome studies is acknowledged (page 4, line 120). Being so, we note (page 6, line 200) that to date most taxa studies of the vaginal microbiome have only focused on abundant species. Yet, we do acknowledge that it may be no coincidence that taxa abundance is driven by genetic factors, especially for taxon that confer a health benefit (page 6, line 189)

Vaginal birth, which has already been demonstrated to result in vertical transmission of gut and vaginal microbes from the mother to the infant gut, could also result in the transfer of the most prevalent and abundant bacterial taxon, *L. crispatus*, to the vaginal microbiota of female infants.

RESPONSE: This is an interesting point made by the reviewer and vertical transmission could explain, in part, the predominance of *L. crispatus* in European American samples. Yet, the twin study design employed provides a natural control for such environmental exposures shared by both members of a twin pair. Since members of both MZ and DZ twin pairs would experience this exposure equally it would not provide an explanation for the difference in intra-class correlations (covariance structure) we observe for *L. crispatus* which is higher in MZs versus DZs in our cohort. This explanation has been added on page 8, line 247.

2. Methods description

The authors should include more basic methods information into the main manuscript (l. 98), such as the type of quality control screening and the type of sequence data (16S rRNA amplicon not metagenomic sequence reads).

The authors should also mention that species assignments can be achieved with 16S rRNA amplicon sequence data for the vaginal microbiota and its species (can they?). - This is controversial, at least for other body sites and taxa!

RESPONSE: Classification of reads to the species level is not controversial for vaginal microbiome data. We have previously published a detailed description of this analysis pipeline (PMID: 23282177), and reporting vaginal microbiome results at the species level is common and frequently expected. With this approach, in cases where reads cannot be assigned without ambiguity to a single species based on the V1-V3 region of the 16S rRNA gene sequence, reads are assigned to a cluster of closely related taxa. We have shown that ~95% of reads were assigned at the species level or a species-level cluster for a subset of 1,017 samples from the

VaHMP (PMID: 23282177) using protocols for sampling, DNA extraction, 16S rRNA sequencing and analysis that were identical to those in this study.

Other comments:

1. "Women of African ancestry tend to harbor more complex vaginal microbiomes that are often associated with bacterial vaginosis, higher susceptibility to sexually transmitted diseases, pelvic inflammatory disease, and premature birth" (l. 56) - The authors should split this sentence and rephrase. It might be technically true, but it sounds terrible to associate African ancestry with bacterial vaginosis, sexually transmitted diseases, premature birth, etc.

RESPONSE: We agree with the reviewer's assessment and this sentence has been changed by removing the reference to women of African ancestry. In addition, throughout the manuscript, we had originally indicated that women in this study self-identified as either being of African or European ancestry, yet, since we do not estimate ancestral background by the use of ancestrally informative genetic markers we have reconsidered this description. Thus, while women may identify with an ancestral origin they are now referred to, as we think more accurately, as either African American or European American women.

2. Throughout the manuscript (abstract + main text): "There were 322 twins of self-identified African (44 pairs) or European ancestry (122 pairs)."  $2 \times (122 + 44) = 332$

RESPONSE: This is correctly pointed out by the reviewer and our typo has been changed throughout the manuscript.

3. "Our study has not identified any specific genetic loci that contribute to the heritability of *L. crispatus*." (l. 152) - This is misleading, as the analysis of host genetics was not part of the study.

RESPONSE: This was only a clarification that we had not measured individual loci but we have removed this statement to avoid any confusion.

4. "...published research... HAS largely focused..." (l. 164)

RESPONSE: Changed as suggested.

5. Table 1

- explain numbers in parentheses

- "Pregnant status" - Should the groups of "Not sure" and "Unknown" be combined?

RESPONSE: The number in parentheses in Table 1 have now been labeled as percentages. The `Unknown` category has been clarified as being missing data.

Reviewer #2 (Remarks to the Author):

The authors tackle the interesting question of whether features of the vaginal microbiome are heritable. They conduct a twin study with twins of European ancestry and twins of African ancestry. Ancestry was self-reported. They then looked at three common vaginal taxa and asked if there was evidence of heritability and find that *L. crispatus* is heritable among women of European ancestry. I detail some concerns about the approach and conclusions below.

Doesn't age also influence vaginal microbiome composition? The subjects are 18 to 78 years old. Could this influence results?

RESPONSE: This is an important observation from the reviewer. Since it is known that the vaginal microbiome composition is likely subject to change due to the increase in post-menopausal pH levels we performed a sensitivity analysis by omitting women whose age was higher than the average age of menopause onset (51 years old). This also addresses, in part, the critical lack of control for menopause status in the Korean twin and family study. Our follow-up results indicated a negligible change in heritability estimates and these are now listed in Supplemental Table 3 and reported on page 4, line 128.

How strong is the heritability score? I see that it's similar to something found in a Korean study (were the methods the same?) but I didn't get a sense from the paper of either how strong or how general we expect this observation to be.

RESPONSE: The strength of the heritability estimated for *L. crispatus* in European Americans was reported at 34.7% (Abstract and page 4, line 123). As the reviewer states this estimate is similar to the value obtained from samples of Korean women (ie., 36.9%) which provides for a consistent estimate across at least two samples and thus our study would agree with expectations. We report on page 5, line 164-168 that the Korean sample used both MZ and DZ twin pairs similar to our study and using a similar sample size as used in our study. We have also included a new paragraph that puts expectations for heritability estimates in context (page 3, lines 83-92).

Could the "lack of heritable taxa" described be simply explained because you only looked at 3 taxa? I was not clear on why the additional taxa that are described, were not used, or, if they were used, why this was not discussed in more detail?

RESPONSE: While we do not use the phrase “lack of heritable taxa” in our manuscript we realize that we did not fully explain the list of taxa reported in Supplemental Tables 2a-2d. We reported on basic statistical summaries of an expanded list of 32 taxa, not amenable to our final statistical genetic models due to the sparseness of vaginal microbiome data (explained on page 4, line 120 and page 3, line 94), for researchers who may benefit from using these estimates in their own research. We have clarified the motivation for providing these supplemental tables on page 8, line 262.

There are no quantitative "host genetic factors" reported, there's just self reported ancestry? This makes me confused as to what this study is adding to prior observations in many other studies that *L. crispatus* is common in women of European descent. "The difference in heritability between self-identified ancestry groups in this case may be due to the increased prevalence of *L. crispatus* dominance among women of European ancestry" this is a statement from the paper so the authors seem to agree.

RESPONSE: The reviewer is correct and on page 5, line 154 we state that, “The difference in heritability between self-identified ancestry groups in this case may be due to the increased prevalence of *L. crispatus* dominance among women of European ancestry. Larger sample sizes will be needed to determine whether this taxon is also heritable among women of African ancestry given that prevalence of *L. crispatus* is less common.” Our previous study (ref: 10 in manuscript) and others have shown that *L. crispatus* is common in European American samples. We provide a further explanation on page 5, line 157 that estimating a more accurate heritability from African American samples is due to lack of covariance coverage as a consequence of the low prevalence of this species.

The heritability model seems to be used out of the box with default parameters but is not clearly described enough for a reader to reproduce the results. This is not acceptable. What are the different ACE model parameters? How were these decided upon? What are the biological rationales for such models? How are they generally used with microbiome data? What are the particular features of microbiome data that make these models appropriate or inappropriate?

RESPONSE: In order to promote replicability of our results we have provided a Github link for the code used to fit and test the models presented in this paper on page 8, line 263. All starting values and parameters are provided in the code to recreate our findings - these are not out of the box values. The twin method has been used in genetic epidemiology studies for over 50 years and is the standard approach for estimating trait heritability on thousands of human traits. We have added a new paragraph to provide this approach more context in the current application (page 3, lines 83-93). In this case we estimated heritabilities derived from measured microbiome proportions within and between twin pairs (page 8, line 245). We provide a rationale and details of the approach on page 8, lines 241-2. More specifically, the biological rationale for the twin model is provided on page 8, lines 241-251. The ACE parameters are described on

page 8, line 251 and their relationship to the MZ and DZ covariance is mathematically defined in the code on lines 45-54.

In sum, I cannot explain clearly what additional contribution this study makes to prior studies that observe different distributions of *L. crispatus* in women of different ancestries. This, and other points above, would need to be addressed for the paper to be acceptable for publication.

RESPONSE: Our previous studies (ref 10 in manuscript) and others have shown the differential distribution of *L. crispatus* between self-identified racial groups. The next logical step that we address in this manuscript is to ask whether this difference might be driven by genetic or environmental sources. Ours is the first twin study to estimate heritability from both African American and European American samples and we find that ~35% of inter-individual differences in the proportion of *L. crispatus* is due to genetic factors in European American samples and may explain, in part, why *L. crispatus* is most prevalent among women of European ancestry in previous vaginal microbiome studies.

REVIEWERS' COMMENTS:

Reviewer #2 (Remarks to the Author):

The authors have addressed my previous comments and increased the clarity of the manuscript.

I still see only limited novelty and relevance in the reported findings, as the presented vaginal microbiota analysis is limited to only three species of which only one shows heritability (*L. crispatus*) and as cited by the authors, this finding has been reported before in a different cohort (Korean women).